# Optical aberrations following implantation of multifocal intraocular lenses: a systematic review and meta-analysis protocol

Christin Henein ![ORCID],[1,2] Clarissa E H Fang ![ORCID],[3] Desta Bokre,[1] Maaz Khan,[1] Ahmed Adan,[1] Yann Bouremel,[1] Mayank A Nanavaty[4,5]

[1]Institute of Ophthalmology, University College London, London, UK
[2]NIHR Moorfields Biomedical Research Centre, London, UK
[3]Manchester Royal Eye Hospital, Manchester, UK
[4]Sussex Eye Hospital, Brighton, UK
[5]Brighton and Sussex Medical School, Brighton, UK

**Correspondence to**
Dr Christin Henein;
c.henein@ucl.ac.uk

## ABSTRACT

**Introduction** Multifocal intraocular lens (IOLs) are used to restore vision at different focal distances. The technology of multifocal IOLs is continually advancing. Optical aberrations a property of lenses that causes spreading of light over a region resulting in a blurred or distorted image. This study aims to systematically review investigator measured and patient reported optical aberrations following implantation of multifocal IOLs during phacoemulsification surgery to treat presbyopia in adults. **Methods and analysis** We will conduct an electronic database search for randomised controlled trials, prospective non-randomised studies, observational studies in Ovid MEDLINE, Ovid EMBASE, Cochrane Central Register of Controlled Trials (CENTRAL), Web of Science, Scopus and ClinicalTrials.gov in March 2021. Eligibility criteria will include quantitative articles written in English and containing data on optical aberrations. Two independent reviewers will screen titles and abstracts and extract data from full texts, reporting outcomes according to Preferred Reporting Items for Systematic Reviews and Meta-Analyses guidelines. Data extraction of key characteristics will be completed using customised forms. Methodological quality will be assessed using Cochrane Handbook 6.2. **Ethics and dissemination** Ethics approval is not required for this review, as it will only include published data. Findings will be published in a peer-reviewed journal and disseminated across ophthalmic networks. We anticipate that the findings of this work will be of interest to multiple stakeholders: people who have undergone cataract surgery, eye health professionals, ophthalmic surgeons, device manufacturers and policy-makers. It will also inform researchers to where there are gaps in evidence and identify areas for future research.
**PROSPERO registration number** CRD42021271050.

## STRENGTHS AND LIMITATIONS OF THIS STUDY

⇒ This systematic review protocol follows the Preferred Reporting Items for Systematic Review and Meta-Analysis Protocols guidelines.
⇒ This systematic review addresses a gap in the current evidence base by providing a comprehensive assessment of reported optical aberrations following new and older generation multifocal intraocular len (IOL).
⇒ There may be a paucity of randomised controlled trials comparing different multifocal IOLs limiting the number of paired wise meta-analysis that can be done.

## INTRODUCTION

Multifocal intraocular lens (IOLs) have multiple focal lengths; if they have two foci, they are called bifocal and if they have three foci, they are called trifocal. This enables the patient with a multifocal IOL (mIOL) to see both objects located at a distance, intermediate distance or near to them. They are three different mechanisms to achieve this: the technology can be refractive, diffractive or combined. Moreover, toric multifocal lens also help to correct the problem of astigmatism.[1]

Traditional monofocal IOLs provide a single point of focus. A newer enhanced monofocals and extended depth-of-focus (EDOF) IOLs which creates a single elongated focal point to enhance the depth of focus. For the purposes of this study, we will assess optical aberrations following the implantation of different types of mIOL and will exclude enhanced monofocal IOL as a well as EDOF IOLs.

It is generally accepted mIOL are good for distance and near focal distances. According to the lens design, they can be refractive, diffractive or combined. The technology of mIOLs is continually advancing. Next-generation IOLs include rotationally asymmetric segmented mIOL, increase in the central area with the aim to improve reading acuity and improved pupil independence.

Optical aberrations a property of lenses that causes spreading of light over a region resulting in a blurred or distorted image. Optical aberrations can present as symptoms

of glare, holes and stars. This symptoms may limit the patient satisfaction achieved with these IOLs and is therefore an important patient-centred outcomes to quantify. Spherical aberrations significantly contribute to quality of retinal image and subjective refraction. Optical aberrations can be reported subjectively using questionnaires or measured objectively by wavefront aberrometry analysis. Contrast sensitivity can be a more useful/objective tool to assess visual function. Recent reviews that compared multifocal with monofocal IOLs reported outcomes on spectacle independence, visual acuity and quality of life.[2 3] To our knowledge, this is the first review comparing different mIOLs with optical aberrations as the primary outcome.

## Review aim

We aim to systematically review investigator measured and patient-reported optical aberrations following implantation of mIOLs during the phacoemulsification surgery to treat presbyopia.

## Methods and analysis

### Inclusion and exclusion criteria
#### Types of studies
We will include randomised controlled trials (RCTs) and non-randomised interventional studies (retrospective or prospective studies). Observational studies will allow us to provide real-world estimates of reported optical aberrations.

#### Types of participants
We will include adults undergoing cataract surgery and desiring correction for anticipated postoperative presbyopia. We will exclude studies with participants with history of laser refractive surgery.

#### Intervention(s)
We will include small incision cataract extraction and multifocal lens implantation. All types of refractive and diffractive multifocal lenses will be included in this review.

#### Comparator(s)
We will include mIOLs or alternative type of mIOL as comparators such as diffractive, refractive and hybrid technologies.

## OUTCOMES
### Primary outcome
► Participant reported optical aberrations such as but not limited to glare and halos.

### Secondary outcomes
► Measured optical aberrations with wavefront analysis.
► Contrast sensitivity as measured by validated test.
► Spectacle independence as determined by the participant or as determined by the investigator.
► Uncorrected near vision acuity.

► Uncorrected distance vision acuity.
► Uncorrected intermediate distance.
► Mean spherical equivalent within ±0.5D.
► Percentage of eyes seeing 20/20 or better for distance.
► Percentage of eyes seeing 20/40 or better for distance.
► Percentage of eyes seeing J2 or better for near vision.
► YAG laser capsulotomy rates.

### Search strategy
In collaboration with an information specialist, a comprehensive search strategy will be performed using a combination of controlled vocabulary and free text terms. Searches will be conducted in Ovid MEDLINE, Ovid EMBASE, Cochrane Central Register of Controlled Trials (CENTRAL), Web of Science, Scopus and ClinicalTrials. gov bibliographic databases. Other relevant sources will be searched such as reference lists of existing systematic reviews of mIOLs. Please see online supplemental file 1 for strategy syntax for Ovid Medline 1946–March 2021 electronic database. We will download references identified in searches (electronic database and additional searches) into Endnote V.X9 reference management software and remove duplicate abstracts.

### Study selection
The screening process will be undertaken using Endnote V.X9. Two review authors will independently assess the titles and abstracts of records and exclude papers that do not meet eligibility criteria. We will obtain the full text of the remaining papers, and at least two authors will assess the papers against the inclusion criteria for the review to determine their eligibility for inclusion. Non-English language papers will be excluded. The review authors will resolve disagreements through mediation with a third reviewer.

## DATA EXTRACTION
Two review authors will extract data independently using Excel. We will prepilot the data extraction template. We will resolve discrepancies by discussion. Two attempts will be made to contact trial investigators for missing data. Data will be directly imported into Review Manager V.5 (RevMan V.5); and the accuracy of the data import will be checked by one author.

We will collect the following information on study characteristics:
► Study design: parallel group RCT/within-person RCT/one or both eyes reported.
► Participants: country, total number of participants, age, sex, inclusion and exclusion criteria.
► Intervention and comparator details: type of mIOL, including number of people (eyes) randomised to each group.
► Primary and secondary outcomes as measured and reported in the trials.
► Length of follow-up.
► Date of publication.

Date mIOL received market approval (U.S Food and Drug Administration premarket approval FDA PMA and European Union Conformitè Europëenne CE mark).

► Sample size.
► Funding and conflicts of interest.
► Trial registration, if available.

## DATA SYNTHESIS

We will pool data where there are at least two studies for a particular type of mIOL reporting the same outcome. We will use a random-effects model in RevMan V.5. But if there are fewer than three trials in a comparison, we will use a fixed-effect model. If there is inconsistency between individual study results such that a pooled result may not be a good summary of the individual trial results—for example, the effects are in different directions or I²>50% and p<0.1—we will not pool the data but will describe the pattern of the individual study results. If there is statistical heterogeneity we may pool the data if all the effect estimates are in the same direction, such that a pooled estimate would seem to provide a good summary of the individual trial results.

We will extract the following data from each included study for intervention and comparator groups separately.

► Number of events and number of participants on which outcome data collected for dichotomous variables.
► Mean, SD and number of participants on which outcome measured for continuous variables.

For multiarm studies, we will use data relevant to our intervention and comparator groups. If two groups contain relevant data we will combine groups using the calculator within RevMan V.5. If an SD is not available, we will use information from confidence intervals and p values, where possible, to estimate it, using the RevMan V.5 calculator.[4]

For the primary outcome, a power calculation will made using metapower package in R (rstudio.com) to calculate the statistical power for meta-analysis based on Cohen's d.[5] We expect to find at least 10 studies with sample sizes of at least 40 participants and we anticipate considerable statistical heterogeneity I²=50%, with an estimated effect size of 0.35. Based on the aforementioned parameters, the estimated power for a fixed effects model is 0.93 and a random effects model is 0.69.

### Assessment of risk of bias in included studies

Two review authors will assess independently the risk of bias using Cochrane's 'Risk of bias' tool for assessing risk of bias in each included study according to the following domains selection bias, performance bias, detection bias, attrition bias and selective outcome reporting bias.[6] We will resolve disagreements by discussion. We will specifically consider and report on the following sources of bias. We will grade each domain as low risk of bias, high risk of bias or unclear (lack of information or uncertainty of potential for bias). We will attempt to contact trial investigators for clarification of parameters graded as 'unclear'.

### Dealing with missing data

If possible, we will conduct an intention-to-treat (ITT) analysis. We will use imputed data if computed by the trial investigators using an appropriate method but will not impute missing data ourselves. If ITT data are not available, we will do an available case analysis. This assumes that data are missing at random. We will assess whether this assumption is reasonable by collecting data from each included trial on the number of participants excluded or lost to follow-up and reasons for lost to follow-up by treatment group, if reported.

### Assessment of heterogeneity and subgroup analysis

We will examine the overall characteristics of the studies, in particular the type of participants and types of interventions, to assess the extent to which the studies are similar enough to make pooling study results sensible. We will look at the forest plots of study results to see how consistent the results of the studies are, in particular looking at the size and direction of effects. We will calculate $I^2$ which is the percentage of the variability in effect estimates that is due to heterogeneity rather than sampling error. We will consider $I^2$ values over 50% to indicate substantial inconsistency but will also consider $\chi^2$ p value. As this may have low power when the number of studies are few we will consider p<0.1 to indicate statistical significance of the $\chi^2$ test. If there are sufficient trials, we will compare the effect of treatment in the following subgroups; diffractive, refractive and hybrid mIOL and year of market approval.

### Sensitivity analysis and assessment of reporting biases

We will examine the impact of excluding studies at high risk of bias in one or more domains. If there are 10 trials or more included in a meta-analysis, we will construct funnel plots and consider tests for asymmetry for assessment of publication bias, according to Chapter 8 of the Cochrane Handbook for Systematic Reviews of Interventions.[7]

### Limitations of this study

Bias such as lack of masking and confounding factors in the studies included will affect the certainty of the estimate of effect in our study. We will aim to mitigate against this by conducting sensitivity analysis by assessing the effect of excluding low quality studies. High heterogeneity among studies would reduce the power of this review. One of the reasons for this could be the use of different tools to measure the prevalence and extent of optical aberrations. Understanding whether the heterogeneity is clinical or statistical will be important and, in some instances, pooling of the data in a meta-analysis may not be appropriate. Publication bias could lead to overestimation of the true effect size, so clinical trial registries will be searched to identify unpublished results where

possible. Furthermore, industry sponsored studies with conflicts of interests among investigators could introduce bias which would need to be evaluated.

**Contributors** MAN conceived the idea for the review. CH, MK, AA and DB drafted and revised the protocol with suggestions from YB, CEHF and MAN who reviewed the protocol and provided feedback on the draft. DB constructed the search.

**Funding** CH receives support from the National Institute for Health Research CL 2020-18-009

**Competing interests** None declared.

**Patient and public involvement** Patients and/or the public were not involved in the design, or conduct, or reporting, or dissemination plans of this research.

**Patient consent for publication** Not applicable.

**Provenance and peer review** Not commissioned; externally peer reviewed.

**ORCID iDs**
Christin Henein http://orcid.org/0000-0002-6972-5355
Clarissa E H Fang http://orcid.org/0000-0002-7266-7117

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
