## [Reviewer comments · BMJ Open]

ARTICLE DETAILS

TITLE (PROVISIONAL)	Optical aberrations following implantation of multifocal intraocular lenses: a systematic review and meta-analysis protocol
AUTHORS	Henein, Christin; Fang, Clarissa E H; Bokre, Desta; Khan, Maaz; Adan, Ahmed; Bouremel, Yann; Nanavaty, Mayank

VERSION 1 – REVIEW

REVIEWER	Penbe, Aysegul Health Science University, Department of Ophthalmology
REVIEW RETURNED	10-Jan-2022

GENERAL COMMENTS	The title and the aim of the study are quite interesting and the study was designed as a protocol, but the results are not about the optical aberrations following implantation of multifocal intraocular lenses as mentioned in the review aims, there is information about the protocol process only. Line 82: The information about the multifocal IOLs in the introduction is insufficient in terms of current approaches and terminologies. It would be appropriate to mention especially edof lens technology. Line 134; J5 threshold is not sufficient for good near vision. The near vision levels from J5 to J1 are so different abilities from each other. Line 136: In secondary outcomes, the intermediate distance should be evaluated.
--

REVIEWER	Grzybowski, A University of Warmia and Mazury, Department of Ophthalmology
REVIEW RETURNED	07-Feb-2022

GENERAL COMMENTS	the Introduction contains some redundant statements that should be excluded.
--

REVIEWER	Rose-Nussbaumer, Jennifer University of California San Francisco, Proctor Foundation
REVIEW RETURNED	20-Jun-2022

GENERAL COMMENTS	Major Comments: Overall, the significance of this paper is limited. This is not a protocol paper for a randomized clinical trial or prospective study, rather it is for a meta-analysis. It is likely to be a challenging to find studies with similar enough outcomes that they can be analyzed together. What will be a minimum number of study subjects to determine if a question is
---

	addressable in the meta-analysis. Will a power calculation be performed to determine the power of the convenience sample? Please add a section about the limitations of the proposed meta-analysis Minor comments: Line 118: adults aged 18 years and above with presbyopia should be re-worded as there are no 18 years old with presbyopia. Maybe “Undergoing cataract surgery and desiring correction for anticipated post-operative presbyopia”
--	---

VERSION 1 – AUTHOR RESPONSE

Reviewer: 1

Dr. Aysegul Penbe, Health Science University

Comments to the Author:

The title and the aim of the study are quite interesting and the study was designed as a protocol, but the results are not about the optical aberrations following implantation of multifocal intraocular lenses as mentioned in the review aims, there is information about the protocol process only.

Thank you

Line 82: The information about the multifocal IOLs in the introduction is insufficient in terms of current approaches and terminologies. It would be appropriate to mention especially edof lens technology.

Introduction has been amended

Line 134; J5 threshold is not sufficient for good near vision. The near vision levels from J5 to J1 are so different abilities from each other.

Outcome has been amended to % of patients near reading vision of J2 or better

Line 136: In secondary outcomes, the intermediate distance should be evaluated.

Uncorrected intermediate distance visual acuity added as secondary outcome.

Reviewer: 2

Dr. A Grzybowski, University of Warmia and Mazury, Poznan City Hospital

Comments to the Author:

the Introduction contains some redundant statements that should be excluded.

Introduction amended

Reviewer: 3

Dr. Jennifer Rose-Nussbaumer, University of California San Francisco

Comments to the Author:

Major Comments:

Overall, the significance of this paper is limited. This is not a protocol paper for a randomized clinical trial or prospective study, rather it is for a meta-analysis.

It is likely to be a challenging to find studies with similar enough outcomes that they can be analyzed together. What will be a minimum number of study subjects to determine if a question is addressable in the meta-analysis. Will a power calculation be performed to determine the power of the convenience sample?

We will pool the results when there is at least two studies for each type of multifocal IOL and conduct meta-analysis when the heterogeneity is within the defined parameters in the protocol. Data synthesis

section has been amended to include this statement. Power calculation has been added to this section.

Please add a section about the limitations of the proposed meta-analysis
A section about the limitations has been added.

Minor comments:

Line 118: adults aged 18 years and above with presbyopia should be re-worded as there are no 18 years old with presbyopia. Maybe "Undergoing cataract surgery and desiring correction for anticipated post-operative presbyopia"
Sentence amended